# Impact of odorants on perception of sweetness by honey bees

**Allyson V. Pel**[1⊚], **Byron N. Van Nest**[2⊚], **Stephanie R. Hathaway**[3], **Susan E. Fahrbach**[4]*

**1** PCOM, Suwanee, Georgia, United States of America, **2** Department of Biological Sciences, University of Manitoba, Winnipeg, Manitoba, Canada, **3** Department of Entomology, Purdue University, West Lafayette, Indiana, United States of America, **4** Department of Biology, Wake Forest University, Winston-Salem, North Carolina, United States of America

⊚ These authors contributed equally to this work.
* fahrbach@wfu.edu

**Data Availability Statement:** The data underlying this article are available in Figshare at https://doi.org/10.6084/m9.figshare.23664006.v1.

## Abstract

Organic volatiles produced by fruits can result in overestimation of sweetness by humans, but it is unknown if a comparable phenomenon occurs in other species. Honey bees collect nectar of varying sweetness at different flowering plants. Bees discriminate sugar concentration and generally prefer higher concentrations; they encounter floral volatiles as they collect nectar, suggesting that they, like humans, could be susceptible to sweetness enhancement by odorant. In this study, limonene, linalool, geraniol, and 6-methyl-5-hepten-2-ol were tested for their ability to alter behaviors related to perception of sweetness by honey bees. Honey bees were tested in the laboratory using proboscis extension response-based assays and in the field using feeder-based assays. In the laboratory assays, 6-methyl-5-hepten-2-ol and geraniol, but neither linalool nor limonene, significantly increased responses to low concentrations of sucrose compared with no odorant conditions in 15-day and 25-day-old adult worker honey bees, but not in 35-day-old bees. Limonene reduced responding in 15-day-old bees, but not in the older bees. There was no odorant-based difference in performance in field assays comparing geraniol and limonene with a no odorant control. The interaction of the tested plant volatiles with sucrose concentration revealed in laboratory testing is therefore unlikely to be a major determinant of nectar choice by honey bees foraging under natural conditions. Because geraniol is a component of honey bee Nasonov gland pheromone as well as a floral volatile, its impact on responses in the laboratory may reflect conveyance of different information than the other odorants tested.

## Introduction

Different types of berry are generally perceived by human consumers to have varying degrees of sweetness, with perception sometimes uncoupled from actual sugar concentration. For example, FoodData Central of the United States Department of Agriculture reports that strawberries contain 4.9 g of sugar per 100 g while blueberries have 9.4 g of sugar per 100 g [1]. In the laboratory, however, human subjects tend to report that strawberries are sweeter than blueberries, despite their lower total sugar content [2].

**Funding:** This research was supported by funds provided to SEF through the Z. Smith Reynolds Foundation Endowment to Wake Forest University. The funders had no role in study design, data collection and analysis, decision to publish, or preparation of the manuscript.

**Competing interests:** The authors have declared that no competing interests exist.

What factors other than concentration of sugar might influence perception of sweetness? The sensory experience of eating a berry combines total sugar content, the specific sugars that are present [3], texture, pH, and, importantly, exposure to volatile odorants via both orthonasal and retronasal pathways [4]. A study of strawberries correlating human consumer evaluations with chemical and physical traits revealed that 30 volatile compounds produced by strawberries enhanced the reported intensity of sweetness [5]. Similar results have been reported for tomatoes [6, 7]. These examples reflect a broader phenomenon that, in humans, selected odorants can enhance perceived sweetness [8, 9]. The degree to which innate effects of specific odorants on the central nervous system contribute to this phenomenon versus a role for prior (learned) flavor-taste associations is difficult to disentangle in human subjects [10].

Pollinators such as honey bees encounter floral odorants as they collect nectar from flowers [11]. Nectars contain different proportions of sucrose, glucose, and fructose and vary widely in total sugar concentration [12]. Honey bees display inter-individual variability in foraging behavior [13, 14], but are sensitive to changes in nectar composition and alter colony-level foraging behavior to exploit more profitable nectar sources [15, 16]. In this study we tested the hypothesis that simultaneous exposure to a floral odorant and a nectar mimic (sucrose solution) can enhance the response of honey bees to low concentrations of sucrose. We also tested the hypothesis that the presence of specific odorants at a sugar water feeder can alter foraging preferences in the field. We are not arguing for comparable sensory mechanisms in humans and honey bees, but rather pursuing this question out of curiosity regarding factors affecting foraging decisions.

Honey bees are good models for the study of this interaction because they forage on an enormous range of flowers that produce an extraordinary diversity of volatile compounds [17]. Insects provide a simpler model than humans because they detect odorants via olfactory sensory neurons embedded in the cuticle of the antennae and adjacent maxillary palps and have no equivalent of a retronasal pathway [18]. Methods for studying the responses of honey bees to sugars in the laboratory and in the field are well-established [19, 20], and honey bees can be reared in controlled environments to control for effects of prior odorant-taste pairings on behavior. However, the vast number and diversity of plant volatiles makes the choice of odorants for testing extremely challenging. The literature on the enhancement of sweetness perception in humans by plant volatiles provided scant guidance, given that these studies tend to either focus on a narrow range of food-associated odorants [9] or apply the tools of biochemical analysis to produce comprehensive lists of plant emissions [7]. The odorants selected for use in this study are widely produced by plants and likely to be commonly encountered by foraging honey bees. To our knowledge, despite an extensive prior literature on the ability of honey bees to learn that floral odorants predict a food reward [21], there is no demonstration of modulation of perception of sweetness in an insect by these or any other odorants.

## Materials and methods

### Honey bees

Adult worker honey bees (*Apis mellifera ligustica)* derived from naturally mated queens were collected from the research apiary at Wake Forest University in Winston-Salem, North Carolina, USA or from remote hives maintained 8 km away within the unincorporated community of Pfafftown, North Carolina, USA. The Office of Laboratory Animal Welfare of the United States Public Health Service does not regulate research on invertebrate animals [22]. The honey bees used in these studies were maintained in hives with normal demography according to standard North American apicultural practices. Honey bees were returned (or, in the case of

field studies, returned of their own volition) to their hives at the end of each day of testing. Each honey bee used in this study experienced only a single day of testing.

## Odorants

The following odorants were purchased from Sigma-Aldrich Chemical Company (St. Louis, MO, USA): (±)-linalool, CAS 78-70-6; 6-methyl-5-hepten-2-ol, CAS 1569-60-4; geraniol, CAS 106-24-1; and (+)-limonene, CAS 5989-27-5. These odorants have been used in prior studies of olfactory-mediated behaviors in honey bees. Linalool is a monoterpene alcohol produced by many flowers [23]. It is a common component of floral scent [24] and imparts sweetness to ripe tomatoes [25]. Linalool-rich lavender is typically included on lists of flowers considered attractive to honey bees and other insect pollinators [26]. 6-methyl-5-hepten-2-ol (6MH) is an unsaturated methylated ketone produced as a volatile by 50% of all studied flowering plants [24]. Geraniol, another common plant volatile, is a terpene alcohol produced by diverse plant families including the Ericaceae, Orchidaceae, and Rosaceae [24]. Geraniol is also a component of a pheromone produced by the abdominal Nasonov gland of worker honey bees [27, 28]. The Nasonov gland pheromone is used by honey bees to orient foragers to the hive entrance and to signal the presence of water or unscented food sources such as sugar syrup feeders [29, 30]. Limonene, a cyclic monoterpene that occurs naturally in the peels of citrus fruit, was included as a potential neutral odorant because there is no direct prior evidence that its presence enhances perception of sweetness, although it is a common component in floral scents and is structurally similar to linalool [24]. In contrast to the other compounds tested, limonene is sometimes used as an insect repellent [31], although in commercial melon production floral production of limonene was found to be correlated with attractiveness to honey bees [32].

## Sucrose solutions

Sucrose solutions were prepared fresh as needed by dissolving sucrose (Sigma-Aldrich Chemical Company St. Louis, MO, USA, > 99.5% purity) in deionized water.

## Laboratory assay for sucrose sensitivity

Approximately 2 weeks before each testing day, frames filled with cells of capped worker brood (late pupal stage, determined by inspection of eye pigmentation) were brought from our remote hives into the laboratory. Frames were held in a cage inside an incubator (Percival, Perry, IA, USA) at 33˚C. Bees that had emerged from their cells (signaling completion of metamorphosis) were collected daily and marked on the thorax with a dot of paint (Testors, Rockford, IL, USA) in a color specific for that day. Marked bees were then added to a hive in the campus apiary that otherwise contained unmarked bees.

Laboratory assays for responses to varying concentrations of sucrose used 15, 25, and 35-day-old adult worker honey bees as subjects. Fifteen-day-old bees are unlikely to have experience foraging outside the hive; 25-day-old workers are in the prime of their foraging careers; 35-day-old workers are likely highly experienced foragers nearing the end of their lives. On test days, 20–30 bees of known age (indicated by paint mark) were collected by opening the hive and inspecting the frames. Paint-marked target bees were transferred to a cage for transport to the laboratory. In the laboratory, cages were placed on ice until the bees were temporarily immobilized. Bees were then harnessed into individual cylinders, hand-fed to satiation with sucrose syrup (2.9 mol L$^{-1}$) and held in an incubator for a pre-test food-deprivation period of 6 h following established protocols [33, 34]. It is important to control feeding history because impact of recent feeding and foraging on responses to sucrose has been demonstrated in

multiple prior studies [35–38]. The high concentration of sugar was used to ensure that, despite being manipulated for harnessing, each of our subjects were physically capable of responding to a stimulus with proboscis extension. This is similar to the use of honey as a positive control in other studies of sucrose sensitivity in honey bees [39].

At the end of the food-deprivation period, harnessed bees were placed in a test box in the center of a standard laboratory fume hood. Approximately 2 min prior to presenting the sucrose solutions, 1 mL of the odorant being tested was applied to a piece of Whatman No. 4 filter paper positioned behind the rack holding the harnessed bees. The concentrations of sucrose used for these tests ranged from 0.3 mol L$^{-1}$ to 1.5 mol L$^{-1}$. They are designated on figures in terms of their equivalent w/v solutions: 10% (0.3 mol L$^{-1}$); 20% (0.6 mol L$^{-1}$); 30% (0.9 mol L$^{-1}$); 40% (1.2 mol L$^{-1}$); and 50% (1.5 mol L$^{-1}$).

A worker honey bee extends its proboscis when the antennae are touched with a drop of a solution the bee perceives as containing sucrose [40, 41]. This unlearned proboscis extension response (PER) reflects stimulation of gustatory receptors on the antennae, including receptors that respond to sucrose [42–45]. This method of testing for sucrose sensitivity is standard in the field of honey bee behavioral research [19, 20] although there are many minor variations (for example, in the scoring system used to report results and the number and concentration of sucrose solutions tested). The distinctive feature of our procedure was to test sucrose sensitivity in the presence of specific floral odorants versus the ambient environment of the laboratory.

Prior to testing, each honey bee was touched on the antennae with a drop of water. Subjects that responded to water alone with a proboscis extension were excluded from testing. Testing then proceeded as follows: in turn, the antennae of each bee were touched (using a separate wooden applicator stick per sucrose concentration) with an ascending series of sucrose concentrations. Each sucrose presentation was separated by a presentation of water to ensure that the subjects responded to the gustatory stimulus presented rather than the mechanosensory stimulus of the light touch to the antennae or as a result of inadvertent learning by contact of the proboscis with the antenna [46–49]. The presence or absence of a fully extended proboscis was recorded for each sucrose presentation. A sucrose response score, calculated for each bee, was defined as the sum of the number of responses to sucrose presentations for that odorant treatment. A score of 5 indicates that a subject responded to all presentations of sucrose. A score of 0 indicates that a subject responded to no presentations of sucrose; these non-responders were not included in subsequent analyses. Subjects that responded to every presentation of water were also excluded. Subjects were not permitted to consume any of the solutions presented. Two minutes elapsed between each stimulus presentation. Table 1 shows the distribution of subjects by odorant condition (no odorant, limonene, linalool, geraniol, and 6MH) and by age (15, 25, and 35 days). Completion of testing required 12 weeks, and all odorant conditions and at least two age groups were included in each week's test schedule.

Shapiro-Wilk tests and inspection of Q-Q plots indicated that the distributions of the sucrose response scores were significantly non-normal, despite the large sample sizes. Consequently, they were analyzed using a Kruskal-Wallis Multiple Comparisons test and when applicable, *post hoc* analyses with Dunn's Test for pairwise multiple comparisons based on rank sums. Developmental trends within odorant conditions were analyzed using Mann-Whitney U tests, and chi-square tests were used to evaluate sucrose concentration dependent responses by odorant condition. Statistics were analyzed using R 3.3.2 [50] and GraphPad Prism version 9.5.1 for Windows (GraphPad Software, San Diego, California USA, www.graphpad.com), both of which return approximate p-values based on the chi-square distribution. Alpha levels to determine significance were set at α = 0.05. Figures were initially prepared

**Table 1. Distribution of subjects by odorant condition.**

| Treatment | 15-day-olds | | | 25-day-olds | | | 35-day-olds | | |
|---|---|---|---|---|---|---|---|---|---|
| | N | Mean (± sem) | Median | N | Mean (± sem) | Median | N | Mean (± sem) | Median |
| No odorant | 122 | 3.7 (0.14) | 4 | 90 | 3.6 (0.15) | 4 | 93 | 4.1 (0.13) | 5 |
| Limonene | 78 | 3.2 (0.19) | 4 | 141 | 3.9 (0.12) | 4 | 65 | 4.0 (0.17) | 5 |
| Linalool | 64 | 4.0 (0.19) | 5 | 91 | 4.0 (1.5) | 5 | 52 | 4.1 (0.19) | 5 |
| Geraniol | 81 | 4.0 (0.16) | 5 | 85 | 4.3 (0.14) | 5 | 57 | 4.3 (0.16) | 5 |
| 6MH | 36 | 4.2 (0.20) | 5 | 76 | 4.1 (0.15) | 5 | 64 | 4.1 (0.18) | 5 |

Summary of number of positive responses (full proboscis extensions) to sucrose (out of a possible maximum of five) for honey bees tested under different odorant conditions.

in GraphPad Prism and customized using Adobe Illustrator 2023 (https://adobe.com/products/illustrator).

### Field assay of gustatory acceptance

Gustatory acceptance is a measure of sucrose sensitivity and persistence displayed by foraging honey bees in the field. Acceptance is scored by identifying the lowest concentration of sucrose that induces an individual honey bee forager to return to a feeder. We adapted the method of Mujagic and Erber [51] to assess the impact of floral odorants on the behavior of free-flying bees. To initiate these studies, foragers residing in a hive on the Wake Forest University campus were trained to visit a feeder 100 m from the hive following the incremental method of Van Nest and Moore [52]. The feeder was a 96-well plate [52]. Each well was initially stocked with approximately 375 µL of 2 mol $L^{-1}$ sucrose solution. When a forager visits such a feeder, she probes the solution with her proboscis, mouthparts, and antennae. If she decides to ingest the solution, she will fill her crop, return to the hive to offload the solution, and then return to the feeder to collect more solution. If the solution is not suitable, the bee will not return to the feeder [51].

Training bees to the location of the feeder required approximately 8 h. We were able to train only once at the start of our field studies because subsequent data collection occurred daily, allowing the previous day's foragers to recall the location of the feeder and to recruit additional subjects. We differentiated between foragers used to recruit other honey bees to the feeder and experimental subjects by placing a small dot of gold paint on the dorsal thorax of each bee that fed from the feeder during the training period. Therefore, every bee that visited the feeder had a gold dot or, on a data collection day, was initially unmarked and received a unique combination of different color paint dots on the thorax to indicate their status. Recruitment of test bees started at 0900 h with a 1.5 mol $L^{-1}$ sucrose solution in the feeder (no odorant present) to incentivize previously trained bees to recruit new foragers. This solution was offered until unpainted foragers appeared at the feeder. The odorant was then introduced by placing a clean, freshly filled 96-well plate on top of a piece of filter paper treated with 1 mL of the odorant being tested. The concentration of sucrose remained at 1.5 mol $L^{-1}$ until six focal bees were recruited. The 1.5 mol $L^{-1}$ sucrose solution (50% w/v) feeder was then replaced with one offering a 0.9 mol $L^{-1}$ sucrose solution (30% w/v) and an additional 1 mL of odorant was added to the filter paper. Testing continued with this solution for 5 min while the total number of visits to the feeder made by focal bees was recorded. After 5 min, the 0.9 mol $L^{-1}$ sucrose solution feeder was replaced with a 0.3 mol $L^{-1}$ sucrose solution (10% w/v) feeder. Feeder visits were recorded for 10 min, after which the sucrose concentration was dropped to 0.09 mol $L^{-1}$

sucrose (3% w/v), also for 10 min. This process continued until the concentration of sucrose reached zero (the feeder contained water) or until none of the focal bees returned for the full duration of a concentration presentation. The full sequence of sucrose solutions offered was: 1.5 mol L$^{-1}$ (for the duration of painting focal bees), 0.9 mol L$^{-1}$ (5 min), 0.3 mol L$^{-1}$ (10 min), 0.09 mol L$^{-1}$ (10 min), 0.03 mol L$^{-1}$ (15 min), 0.009 mol L$^{-1}$ (15 min), 0.003 mol L$^{-1}$ (15 min), and then water (10 min). A gustatory acceptance score was defined per bee by identifying the lowest concentration to which the bee returned. For example, if a bee returned to all concentrations until 0.09 mol L$^{-1}$ sucrose, returned once after the switch to 0.03 mol L$^{-1}$ sucrose, and never returned thereafter, her score was 0.09 mol L$^{-1}$. If more than 6 foragers were present at the feeder during data collection, superfluous bees were detained in glass vials and held on ice until the end of that day's testing. At this time, the concentration of sucrose was switched back to 1.5 mol L$^{-1}$, and the sequestered bees were allowed to visit the feeder to incentivize return the following day.

Gustatory acceptance field assays were conducted using no odorant, geraniol, and limonene; geraniol was selected because of its positive effect in the laboratory and limonene was selected for its neutral or potentially repellant effect. Bees that did not return when the sucrose concentration was dropped below the recruiting concentration of 1.5 mol L$^{-1}$ and bees that returned to water were eliminated from the analysis. Data were compiled by odorant treatment: no odorant (n = 64); geraniol (n = 92); and limonene (n = 46). We then assigned bees to one of the following three gustatory acceptance categories: Low (0.1%, 0.3%), Medium (1%, 3%), and High (10%, 30%). The resulting distributions were analyzed using the Chi-square analysis in GraphPad Prism version 9.5.1 for Windows (GraphPad Software, San Diego, California USA, www.graphpad.com). Alpha levels to determine significance were set at α = 0.05. Figures were initially prepared in GraphPad Prism and customized using Adobe Illustrator 2023 (https://adobe.com/products/illustrator).

## Drinking duration assay

The drinking duration assay is a potentially more sensitive measure of sweetness perception than laboratory analyses based on proboscis extension or field assays of acceptance of low sucrose concentrations at a feeder [53]. The drinking duration assay procedure was similar to that described for the gustatory acceptance assay. New focal bees were given a unique paint code when they visited a feeder offering a 0.9 mol L$^{-1}$ sucrose (30%). Six focal bees were painted at a time. The concentration of sucrose used in drinking duration testing to 0.3 mol L$^{-1}$ (10%). Data collection occurred after a switch from 0.9 mol L$^{-1}$ (bait) to 0.3 mol L$^{-1}$ (test) sucrose. The drinking duration of each focal bee was recorded in sec during a 15-min observation period using a stopwatch (S1 Video). A bee was identified as drinking if her head was down, her proboscis was extended into a well of the feeder, and her abdomen was expanding and contracting (pumping). Because viscosity varies with sugar concentration and honey bees can change their mode of feeding (lapping or sucking) in response to different nectars [54], it is important that comparisons of drinking duration not be made across multiple concentrations.

Drinking duration assays were conducted using no odorant (n = 34), geraniol (n = 29), and limonene (n = 31). Individual drinking durations were compiled by odorant. Shapiro-Wilk tests and inspection of Q-Q plots indicated that the distribution of the drinking duration data was significantly non-normal. They were analyzed using a Kruskal-Wallis Multiple Comparisons test. *Post hoc* pairwise comparisons were made using Dunn's Test. Statistics were analyzed using GraphPad Prism version 9.5.1 for Windows (GraphPad Software, San Diego, California USA, www.graphpad.com). Alpha levels to determine significance were set at α = 0.05. Figures

were initially prepared in GraphPad Prism and customized using Adobe Illustrator 2023 (https://adobe.com/products/illustrator).

### Ambient testing conditions

All field tests were conducted on sunny days during which the ambient temperature ranged between a low of 25˚C to a high of 29˚C. Although it is possible that temperature affected viscosity of the offered sugar solutions and the physiology of the honey bees, changes over this temperature range are likely to be negligible.

## Results

### Effect of odorants on gustatory responses to sucrose in the laboratory

A Kruskal-Wallis test was performed on the summed sucrose response scores of the five groups (no odorant, limonene, linalool, geraniol, and 6MH) for each age group (Fig 1). For 15-day-old honey bees the differences of the rank totals of 186 (no odorant), 156 (limonene), 205 (linalool), 208 (geraniol), and 219 (6MH) were significant, Kruskal-Wallis H (4, 381) = 15, p < 0.01. Focused *post hoc* comparisons conducted with Dunn's test compared each of the floral odorant testing conditions with the no odorant group (Fig 1A). The difference between the no odorant and limonene group was statistically significant (p < 0.05), indicating that the presence of limonene reduced responding to sucrose. No other comparisons were significant. Bees exposed to linalool, geraniol, and 6MH at the time of sucrose response testing tended to have higher scores in the assay than bees exposed to limonene or no odorant, but this trend was not statistically significant.

For the 25-day-old honey bees, the differences of the rank totals of 210 (no odorant), 232 (limonene), 246 (linalool), 275 (geraniol), and 257 (6MH) were significant, Kruskal-Wallis H (4, 483) = 13, p < 0.01. Focused *post hoc* comparisons using Dunn's test indicated that the

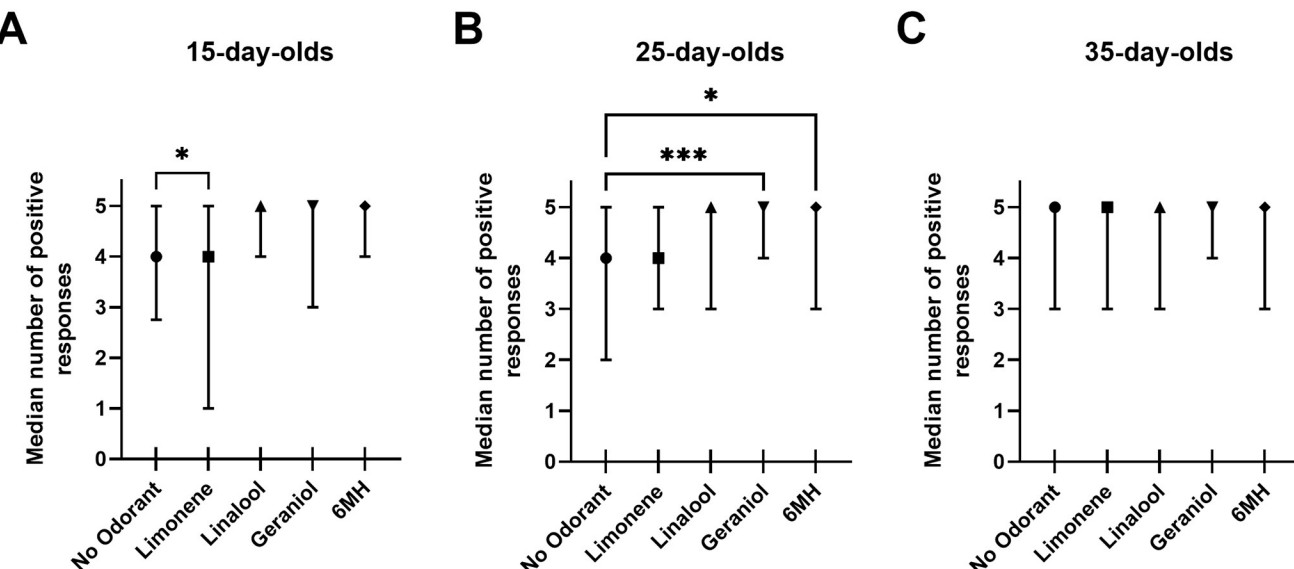

**Fig 1. Proboscis extension responses to sucrose solutions in the presence of no odorant, limonene, linalool, geraniol, or 6MH.** Data are summarized as sucrose response scores (maximum score, 5). Plots show median and interquartile ranges for honey bees tested at different ages. Results of *post hoc* comparisons (Dunn's tests): *, p < 0.05; ***, p < 0.001. Sample sizes given in Table 1. (A) 15-day-old worker honey bees. (B) 25-day-old worker honey bees. (C) 35-day-old worker honey bees.

differences between the no odorant and geraniol groups ($p < 0.001$) and the no odorant and 6MH groups ($p < 0.02$) were significant (Fig 1B). Both geraniol and 6 MH increased responses to sucrose when present. The other comparisons were not significant. There was no effect of the presence of odorant during sucrose response testing in 35-day-old bees, Kruskal-Wallis H $(4, 331) = 1.4$, $p = 0.41$ (Fig 1C).

We asked if there were developmental changes in responses to sucrose by analyzing the data in terms of responses to specific odorants at each of the ages tested using the Mann-Whitney U test for pairwise comparisons of the different age groups (corrected for ties). Thirty-five-day-old bees had higher sucrose response scores in the no odorant (control) condition than 15-day-old ($p < 0.03$) and 25-day-old bees ($p < 0.02$), which did not differ. An inhibitory effect of limonene on responses to sucrose was evident only in the 15-day-old group but not in older bees (15-day-old vs. 25-day-old, $p < 0.003$; 15-day-old vs. 35-day-old, $p < 0.002$). All other developmental comparisons returned non-significant results.

We also examined the incidence of PER by sucrose concentration (Fig 2). In 15-day-old bees, analysis of responses by sucrose concentration revealed no odorant associated differences (Fig 2A). In 25-day-old honey bees (Fig 2B), chi-square tests indicated that the presence of 6MH and geraniol significantly increased the incidence of PER relative to the no odorant condition at the lowest concentration of sucrose tested in this assay (10%); 6MH also increased

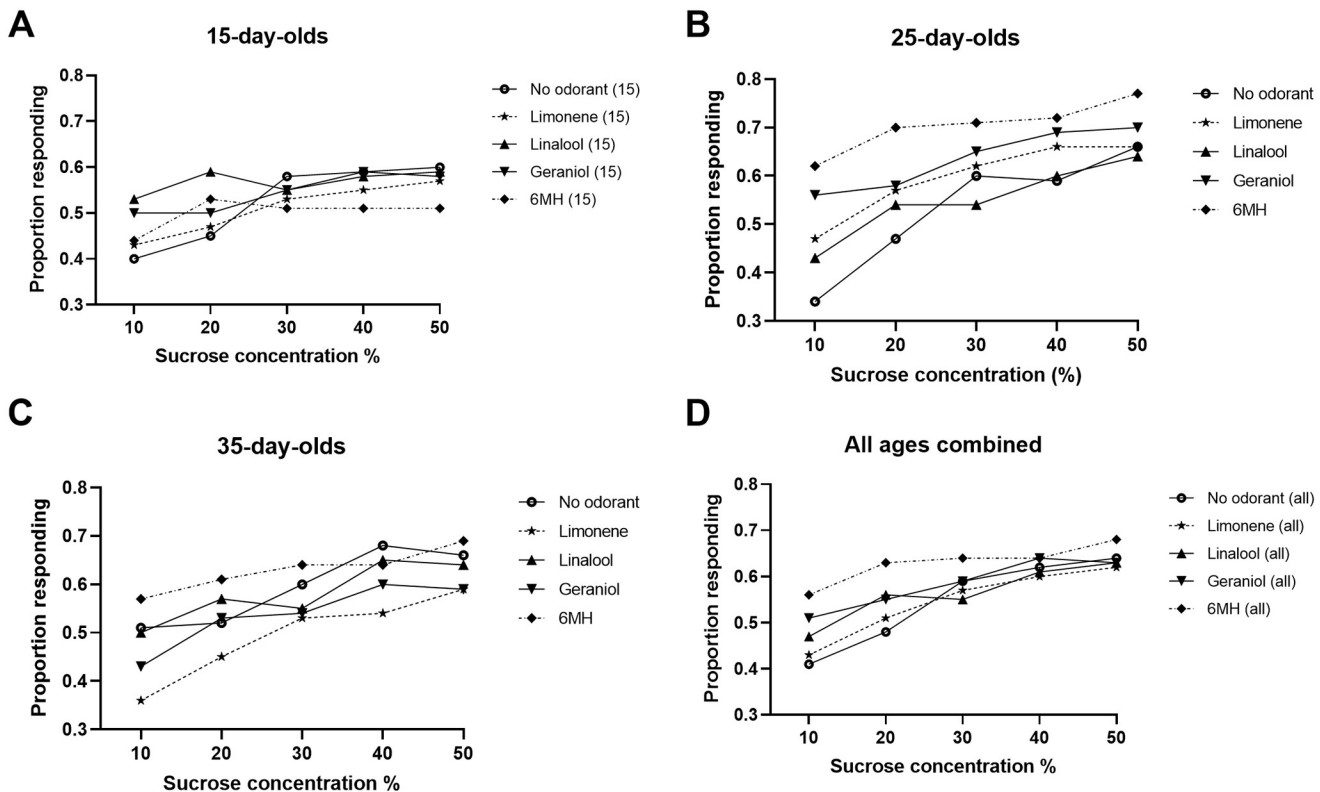

**Fig 2. Concentration-dependent effect of odorants on proportion of honey bees responding to sucrose solutions in the presence of no odorant, limonene, linalool, geraniol, or 6MH.** (A) 15-day-old worker honey bees. No significant differences were observed. (B) 25-day-old worker honey bees. Significant differences were observed at 10% sucrose between 6MH and no odorant ($p < 0.01$); geraniol and no odorant ($p < 0.02$); and 6MH and linalool ($p < 0.05$). The difference between 6MH and no odorant was also significant at 20% sucrose ($p < 0.02$). (C) 35-day-old worker honey bees. There was a significant difference between 6MH and limonene at 10% sucrose only ($p < 0.05$). (D) All ages combined. No significant differences. Sample sizes given in Table 1.

responding relative to linalool at this concentration. At the next highest sucrose concentration (20%), 6MH also significantly increased the incidence of PER in 25-day-old bees relative to the no odorant condition. For 35-day-old honey bees (Fig 2C), the only concentration-dependent difference in proportion of tested bees responding with PER was the comparison between 6MH and limonene at 10% sucrose. Combining all ages in a single analysis (Fig 2D) masked the age-specific differences evident in 25-day-olds and 35-day-olds.

### Field assays of the impact of odorants on responses to sucrose

No effect of odorant was found on the gustatory acceptance score measured in the field for the no odorant, limonene, and geraniol conditions tested [Fig 3: $X^2$ (2, n = 202) = 0.51, p = 0.97)]. No effect of the presence of an odorant was found on drinking duration in the field for the no odorant, limonene, and geraniol conditions [Fig 4: Kruskal-Wallis H (2, 94) = 0.67, p = 0.71].

## Discussion

Nectar is a sweet liquid produced by plants to entice pollinators [11] that also contains non-sugar components including amino acids and plant secondary metabolites [12]. Honey bees foraging under natural conditions encounter a wide range of nectar chemistries, and the assumption that consumption preferences are determined exclusively by sensitivity to sugar concentration is undoubtedly simplistic. Many factors, including genotype [19, 41, 55, 56] and the ovarian status of workers [57] interact in unknown proportions to guide individual foraging decisions. The multiplicity of inputs that fine tune foraging behavior, however, should not obscure the importance of sugar in nectar as an energy source and as a behavior-reinforcing reward, and there is substantial evidence that honey bees convey information regarding sucrose concentration in dances used to recruit other foragers [58, 59]. Sucrose sensitivity measured in the laboratory has been shown to predict the subsequent likelihood of foraging for pollen or nectar [19, 41, 55].

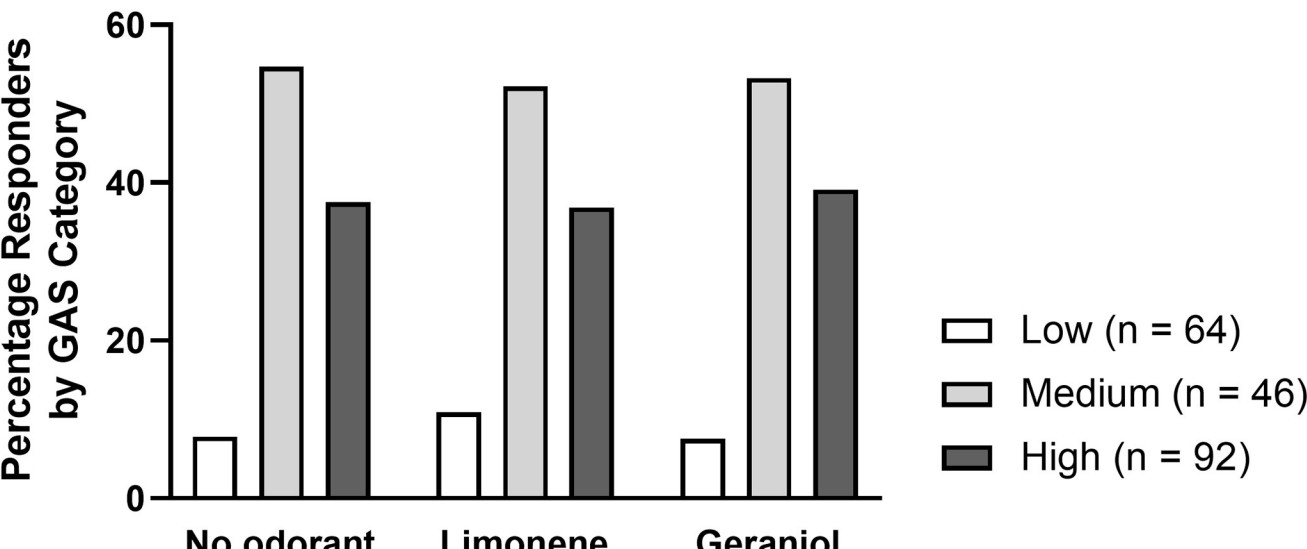

**Fig 3. No impact of limonene or geraniol on gustatory acceptance in the field.** GAS = gustatory acceptance score. Honey bees in the Low category have the greatest sucrose sensitivity, as they will return to feeders offering the lowest concentrations of sucrose. Low concentrations were 0.1% and 0.3% sucrose; Medium concentrations were 1% and 3% sucrose; High concentrations were 10% and 30% sucrose).

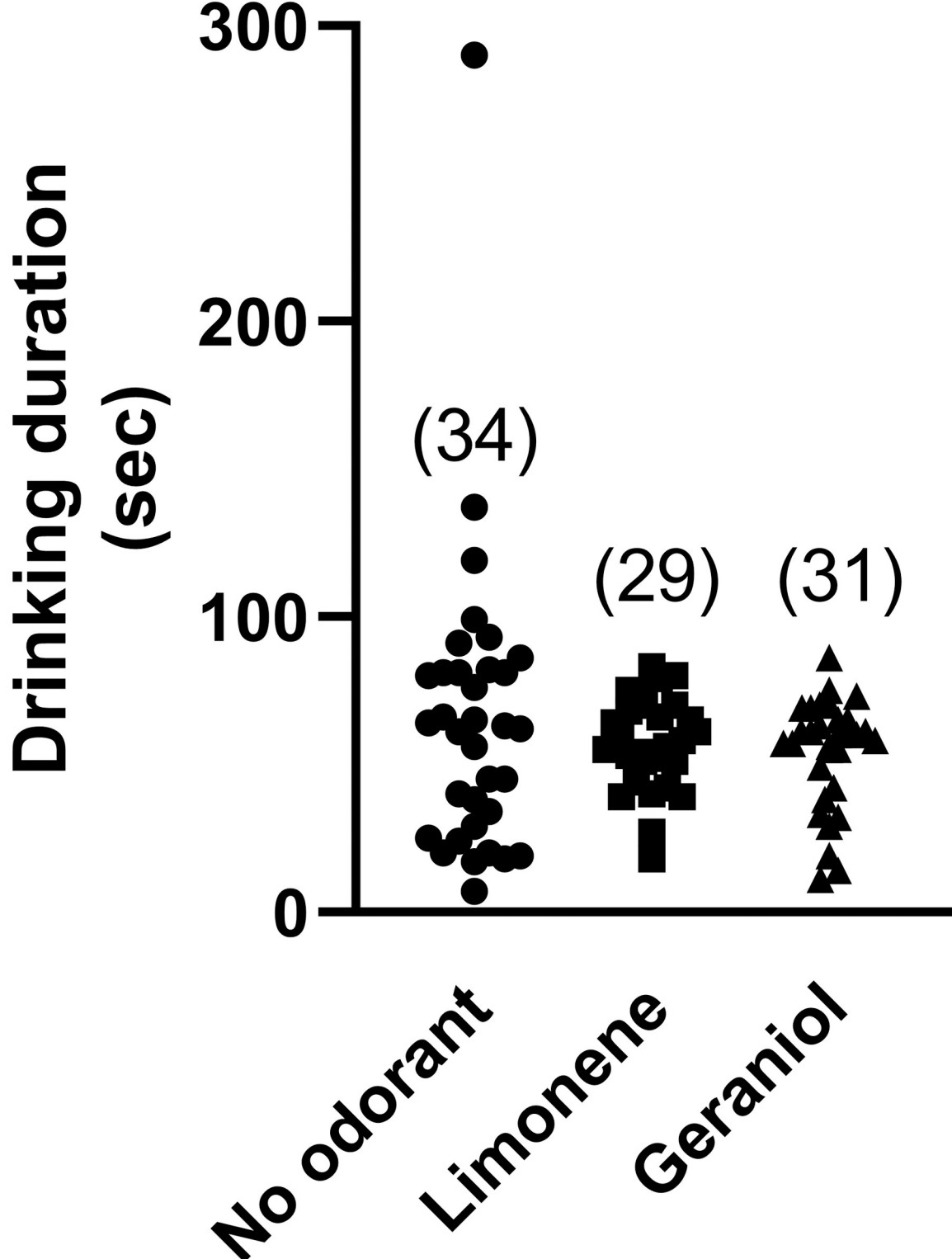

**Fig 4. No impact of limonene or geraniol on cumulative drinking duration (seconds) during a 15-min field observation.** Testing was conducted using a sucrose concentration of 0.3 mol $L^{-1}$ (10%). Sample sizes given in parentheses. Each point represents an individual forager.

Tests of sucrose sensitivity in 15-day-old bees in the laboratory revealed that none of the tested odorants increased responding to sucrose solutions in comparison with the no odorant condition, but that limonene significantly decreased responding in this group. In a honey bee colony with a normal demographic distribution of workers, the transition from performing within hive tasks to foraging occurs on average around 21 days of adult life [60]. The 15-day-old worker bees we tested therefore most likely had not foraged or had only very limited foraging experience prior to collection for testing. Such workers primarily have experience with odorants within the hive, including odorants from pollen and nectar sources within the hive's foraging range. They do not have to evaluate the value of different nectars and are unlikely to have learned that a specific floral odorant predicts a sucrose reward. Based on the results from 15-day-olds, we speculate that the enhanced responses in the presence of floral odorants displayed by the older foragers in this study could represent learned associations. If this is indeed the case, it appears that limonene may have a slight unlearned repellant effect that is overcome as a result of foraging experience.

Tests in 25-day-old workers identified geraniol and 6MH as potential sweetness enhancers for honey bees, as subjects in this age group tested under these conditions responded to a wider range of sucrose concentrations than the other age groups we tested. Worker honey bees of this age are highly likely to have accumulated at least several days of foraging experience [60]. The age-based difference in the effect of geraniol and 6MH on GRS scores of 15- and 25-day-old bees potentially reflects the prior experience of the 25-day-olds with these odorants in the context of visiting flowers. This possibility can be controlled for in future experiments by testing honey bees reared under conditions with no or limited exposure to floral odorants.

In contrast to the 15-day and 25-day-olds, there were no odorant effects on responses to sucrose in 35-day old workers. Given that honey bees foraging in summer (the time this testing occurred) typically have at most a 10-day foraging career [61], this group likely constituted highly experienced foragers. But given that honey bees active in summer have a reported life span of no more than 4–5 weeks [62, 63], it is also likely that the subjects in this group were near the end of their lives. How senescence might influence sucrose sensitivity or more generally, the behavior of harnessed bees, is not known. We noted that the 35-day-old bees gave significantly more responses in the no odorant (control) condition than the two younger groups. As a result, it may have been more difficult to detect a subtle enhancement effect, should one exist. Future studies should control for foraging experience as well as chronological age as in previous studies [64] and use additional test solutions with lower sucrose concentrations, which may make enhancement easier to detect.

### Field tests of odorants at sucrose feeders as an alternative to laboratory testing

Previous studies have proposed a disconnect between laboratory and field assays of sucrose sensitivity in honey bees [51, 53] leading us to ask if odorant-based changes in sucrose sensitivity detected in the lab could also be detected in the field. Because geraniol was reliably found to have a potential sweetness-enhancing effect in our laboratory studies and limonene appeared to have a weak repellant or neutral effect, we tested geraniol and limonene versus no added odorant in the field studies. Although the concentrations of sucrose solution that induced foragers to return to the feeder were lower than those that reliably elicited proboscis extension in the laboratory, no odorant-dependent differences were detected in these studies. One possible explanation is that, compared with the other sugars found in nectar (glucose and fructose), honey bees prefer sucrose [65]. A feeder offering a sucrose-only solution may have been perceived as highly valuable, making it difficult to detect an odorant-based enhancement of

preference. This type of explanation invokes a threshold model of forager responses to sucrose–once the concentration of sucrose exceeds some threshold, a forager no longer attends to other cues.

The measurement of drinking duration at a feeder has been proposed as a more sensitive measure of sweetness perception than voluntary return to a feeder [51]. Nectar loads are typically smaller than average crop capacity of 70 μL [11], implying a capacity for foragers to regulate intake based on the quality of a nectar source. Indeed, a previous study did reveal a positive correlation between sucrose concentration and the size of an individual honey bee's crop load [66], suggesting that foragers consume more of a more rewarding resource. We predicted that drinking duration would be greater in the presence of a sweetness enhancing volatile when compared with no-odorant controls. This prediction was not supported, as we found no odorant-dependent differences in drinking duration at a feeder offering a 0.3 mol $L^{-1}$ sucrose. Our field testing was conducted at the end of summer, when naturally occurring floral sources of nectar are becoming scarce, and anecdotal observations at the time indicated that our subjects were enthusiastic about visiting the feeder. Future studies of drinking duration may benefit from using lower concentrations of sucrose and testing at other times in the field season.

## Significance of geraniol

Geraniol is a naturally occurring floral compound but is also produced by the abdominal Nasonov gland of the honey bee. Geraniol is one component of a blend of seven compounds that functions as a general orientation signal. Nasonov pheromone is used to mark the entrance to the hive, sources of water, and unscented sugar syrup feeders; flowers are rarely marked, and the addition of scent to feeders diminishes the emission of pheromone [29, 30, 67]. Consequently, geraniol, unlike the other odorants tested, has a special status in honey bee life. Of the odorants tested, geraniol was one of the two that appeared to function as an enhancer of sweetness, mimicking the effect observed in humans in the presence of only certain plant volatiles. Another possibility, however, is that experienced foragers have previously learned to associate the scent of geraniol produced by flowers at which they have foraged with a profitable source of naturally occurring nectar (we rule out a role for bee-produced Nasonov pheromone in this context because the bees we tested in the laboratory had no prior experience with artificial feeders). This finding strengthens our speculation that the lack of prior foraging experience can explain why the enhancing effect of geraniol was not observed in inexperienced 15-day-old workers. A third possibility is that geraniol generally enhances attention in honey bees, even in the stripped-down environment of the laboratory. A detailed study of antennal movements in response to a panel of odorants in harnessed honey bees revealed that antennal position and the velocity of antennal movements were altered in the presence of geraniol, while linalool had no impact on antennal responses [68].

## Concluding remarks

Honey bees are generalist foragers that pollinate a wide range of flowering plants. Generalist foragers generate competition among species of flowering plants, which compete by developing odorant-based strategies to attract effective pollinators [69]. In this context, we asked if an odorant could alter responses to sucrose concentration, possibly leading pollinators to perceive nectar as sweeter and therefore more profitable, incentivizing increased visits? The present study demonstrated the impact of the presence of a subset of tested odorants on the gustatory responses of adult worker honey bees, but only in laboratory assays of sucrose sensitivity and primarily in experienced foragers.

## Supporting information

**S1 File. Nectar_foragers_visiting_96-well_plate_feeder.mp4.** This short video shows free-flying honey bees visiting a feeder made from a 96-well plate in the field.
(MP4)

## Acknowledgments

The authors acknowledge Wayne Silver and Joost Maier for thoughtful comments on an earlier version of this manuscript and Aadam Haque, Martina Lammel Knebl, and Seth Tennant for assistance with behavioral testing.

## Author Contributions

**Conceptualization:** Allyson V. Pel, Byron N. Van Nest, Susan E. Fahrbach.

**Data curation:** Allyson V. Pel, Byron N. Van Nest, Susan E. Fahrbach.

**Investigation:** Allyson V. Pel, Stephanie R. Hathaway.

**Methodology:** Allyson V. Pel, Stephanie R. Hathaway, Susan E. Fahrbach.

**Resources:** Susan E. Fahrbach.

**Supervision:** Byron N. Van Nest.

**Writing – original draft:** Allyson V. Pel, Stephanie R. Hathaway, Susan E. Fahrbach.

**Writing – review & editing:** Allyson V. Pel, Byron N. Van Nest, Stephanie R. Hathaway, Susan E. Fahrbach.

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
