## [Decision Letter · Decision Letter 0]

2 Oct 2023

PONE-D-23-24418Impact of odorants on perception of sweetness by honey beesPLOS ONE

Dear Dr. Fahrbach,

Thank you for submitting your manuscript to PLOS ONE. After careful consideration, we feel that it has merit but does not fully meet PLOS ONE’s publication criteria as it currently stands. Therefore, we invite you to submit a revised version of the manuscript that addresses the points raised during the review process.

Reviewer #1, in particular, would like to see a more solid justification as to why honeybees' taste perception should be similar to that of humans and what possible role smells might play in the context of foraging behavior. Reviewer #2 also has a few specific questions that should be addressed.

We look forward to receiving your revised manuscript.

Kind regards,

Wolfgang Blenau

Academic Editor

PLOS ONE

Journal Requirements:

Reviewers' comments:

Reviewer's Responses to Questions

**Comments to the Author**

1. Is the manuscript technically sound, and do the data support the conclusions?

Reviewer #1: Yes

Reviewer #2: Yes

2. Has the statistical analysis been performed appropriately and rigorously? 

Reviewer #1: Yes

Reviewer #2: Yes

3. Have the authors made all data underlying the findings in their manuscript fully available?

Reviewer #1: Yes

Reviewer #2: Yes

4. Is the manuscript presented in an intelligible fashion and written in standard English?

Reviewer #1: Yes

Reviewer #2: Yes

5. Review Comments to the Author

Reviewer #1: The starting question of the project was whether some floral odors might positively affect sugar taste perception. For this the authors tested behavioral responses to sucrose in a lab assay and with free flying honey bee workers of different ages. Although the lab assay suggested an effect on the sugar-elicited responses in the lab assay, the odor stimulation did not affect sugar-dependent behaviors in the free flight condition. So, basically the experiments could not demonstrate any effect of odors on the taste perception of sugar.

The experiments are thoroughly done, and the manuscript is well written.

However, while reading the manuscript I thought by myself, why did the authors do that experiment? Why should the taste system of humans and honey bee forager be the same? Honey bees are not food generalist, they are specialists and nectar is quite a simple food regarding its chemical components, in particular when compared to fruits. And nectar does not need additional sweeteners to override possible repellent components as fruits. Furthermore, honey bees forager, which are recruits that exploit a food source till it is finished, are not that picky, and the food selection is done at the colony level. In addition, the smell of flowers, e.g. geraniol, is likely mainly a long-range attractant for honey bees. If there are close range odors that might affect the taste perception of sugars, these might be less volatile components of the floral odor mixture.

To conclude, in my opinion the study lacks a clear and ecological informed rational why the taste perception of honey bees should be similar to that of humans and the possible role of odors in the context of foraging behavior. I think the authors should add at least some thoughts about these questions. In addition, they could shorten the discussion to avoid unnecessary hand-waving discussions why they did not find any effect of odors on taste perception in honey bees.

Reviewer #2: This paper asks a very original question, which based on data from humans, and its potential for plant-pollinator interactions, definitely should be studied. The results are not extremely strong, but do reveal some statistically significant effects. The authors present these well and discuss them in a critical fashion.

Specific comments:

Lines 51-56. This is an interesting example. Can you add info on the types of sugars in the different berries – are they the same. In principle, a particular food may contain more sugars than another, but mostly of a sugar that is perceived as less sweet than another, and therefore that food will be perceived as less sweet. This would still be due to taste, without a need to invoke also olfaction.

Line 58. Specifically, sugar content is not only sugar concentration but also sugar composition. In honey bees, at least, sucrose is perceived as sweeter than glucose or fructose on their own [this in fact is stated later in the discussion].

Line 142. The sucrose concentration of 2.9 mol L-1 seems extremely high. Wouldn’t it be an almost 100% sucrose solution.

Line 161-170. What was the purpose of touching the antennae with water between each sugar trial? Were these responses recorded? Were these responses considered in any further analyses?

Line 199. There are 96 wells? Are all filled with sucrose solution? What volume in each?

Line 260-263. Ambient temperature would probably also affect imbibing time (both because it affects viscosity and also bee physiology). Could you mention this potential factor and how it was addressed?

Line 341. I suggest adding a brief description of what are the Low, Medium and High labels in the figure (even if it appears in the methods).

Lines 351- 354. I agree. Micronutrients in nectar also greatly affect subjective evaluation of nectar, even when sugar concentrations are equal.

Line 398. This has been studied some, for example: Laloi, D., Gallois, M., Roger, B., Pham-Delegue, M.H., 2001. Changes with age in olfactory conditioning performance of worker honey bees (Apis mellifera). Apidologie 32, 231-242.

I would check from this reference to additional more recent ones.

Line 402. I agree. In fact this is probably one of the main, unfortunate weaknesses of the present study. In sugar sensitivity tests usually lower concentrations are tested.

6. PLOS authors have the option to publish the peer review history of their article (what does this mean?). If published, this will include your full peer review and any attached files.

Reviewer #1: No

Reviewer #2: **Yes: **Sharoni Shafir

---

## [Author Response · Author response to Decision Letter 0]

15 Nov 2023

Response to Reviewer #1

We agree that there is no reason to expect that the taste sensory system of humans and honey bees are the same. In fact, we avoided assertions of analogous structures and pathways and focused on the knowledge base of honey bees feeding on nectar in the field and on sucrose solutions in experimental situations. The study was primarily an exercise in curiosity based on the demonstrated relationship between plant odorants and sweetness perception in humans. It was inspired by a lively conversation at a conference with Professor Linda Bartoshuk (currently of the University of Florida Center for Smell and Taste) on the comparable phenomenon in humans [e.g., Bartoshuk LM & Klee HJ. (2013) Better fruits and vegetables through sensory analysis. Current Biology 23: R374-8.] She directed us to further literature in this area [e.g., Baldwin EA, Goodner K & Plotto A. (2008) Interaction of volatiles, sugars, and acids on perception of tomato aroma and flavor descriptors. Journal of Food Science 73:S294-307.] Subsequent discussions with sensory physiologists at our home institution of Wake Forest University (Wayne Silver, Joost Maier) encouraged us to conduct the experiments reported in the present article. The current project is empirical, not theoretical, although we believe it contributes to the understanding of the multiple determinants of food acceptance in honey bees. We agree that the results are perhaps not surprising but feel that we were careful not to overgeneralize results obtained in the laboratory to claim that floral odorants modify foraging behavior under natural conditions. The results are helpful to us in thinking and designing future experiments on the function of geraniol [(2E)-3,7-dimethyl-2,6-octadien-1-ol] as a signal in honey bee life. We think that these data might be helpful to other researchers interested in the odorant environment experienced by pollinators. 

We have added a comment to the effect that this project was curiosity-driven to the Introduction (Lines 75-77). We do think, however, that it is an oversimplification to assert that all food selection decisions by honey bees are made at the colony level, given that a honey bee colony can exploit multiple food sources at the same time and that modeling suggests decentralized control of foraging decisions [Seeley TD, Camazine S, Sneyd J. (1991) Collective decision-making in honey bees: how colonies choose among nectar sources. Behavioral Ecology & Sociobiology 28:277-290.]We thank Reviewer #1 for the recommendation to shorten the Discussion and have made some cuts in that section (the previous conclusion was deleted; replaced by lines 379-386 which are now 471-478.

Response to Reviewer #2 

Lines 51-56, also 58. This reviewer asked about the sugar composition of the different berries used in human studies. We are interested in this, too! We thought that the use of a single sugar (sucrose) in our studies would be a more rigorous test for odorant-induced changes in behavior but have of course wondered if there is an odorant x sugar interaction in both humans and honey bees. It was reported in a 2019 comparison of the sugar profiles of strawberries and blueberries that the three most abundant sugars in the tested cultivars of strawberries were glucose, fructose, and sucrose, while the three most abundant sugars in blueberries were glucose, fructose, and galactose [Akšić MF, Tosti T, Sredojević M, Milivojević J, Meland M, Natić M. (2019) Comparison of sugar profile between leaves and fruits of blueberry and strawberry cultivars grown in organic and integrated production system. Plants (Basel). 8: 205.] Although we chose not to explore a possible odorant x sugar interaction in our initial study, we now mention this information briefly in the Introduction (Lines 56 and 58-59). 

Line 142: We agree that 2.9 mol L-1 is a high sucrose concentration. We wanted to use a high concentration that almost any honey bee will respond to ensure that each of our subjects, despite being manipulated for harnessing, were physically capable of responding. This is similar to the use of honey as a positive control in other studies of sucrose sensitivity in honey bees [e.g., Siegel AJ, Freedman C, Page RE Jr. (2012) Ovarian control of nectar collection in the honey bee (Apis mellifera). PLoS One 7:e33465.] We chose to use a highly concentrated sucrose solution for this purpose to avoid the variability inherent in honey. This is now discussed briefly in Lines 147-150, and the Siegel et al. (2012) reference has been added.

Line 161-170: The use of water between presentations of sucrose solution of different concentrations was to ensure that the honey bees responded selectively to the gustatory stimulus presented rather than to the mechanosensory stimulus of the light tap on the antenna or as a result of inadvertent learning by contact of the proboscis with the antennae. These data were not analyzed given the lack of any hypothesis regarding the effect of odorants on responses to water, but any subject that responded indiscriminately to every presentation (water and sucrose) was removed from the analysis. We have added a statement to this effect in the Methods. Studies of responses evoked by antennal stimulation vary in whether they include or do not include “blank” trials between tastant presentations [e.g., blanks included: Pankiw T, Page RE Jr. (1999) The effect of genotype, age, sex, and caste on response thresholds to sucrose and foraging behavior of honey bees (Apis mellifera L.). Journal of Comparative Physiology A 185:207-13; Moauro MA, Balbuena MS, Farina WM. (2018) Assessment of appetitive behavior in honey bee dance followers. Frontiers in Behavioral Neuroscience 12:74; Guiraud M, Hotier L, Giurfa M, de Brito Sanchez MG. (2018) Aversive gustatory learning and perception in honey bees. Scientific Reports 8:1343; Bennett MM, Cook CN, Smith BH, Lei H. (2021) Early olfactory, but not gustatory processing, is affected by the selection of heritable cognitive phenotypes in honey bee. Journal of Comparative Physiology A 207:17-26] versus blanks not included: Wang Y, Brent CS, Fennern E, Amdam GV. (2012) Gustatory perception and fat body energy metabolism are jointly affected by vitellogenin and juvenile hormone in honey bees. PLoS Genetics. 8(6):e1002779]. Citations supporting our use of blanks have been added to Lines 170-180. 

Line 199. The type of feeder used in studies of free-flying honey bees is described in detail in a technical paper by co-author Van Nest [Van Nest BN, Moore D. (2018) How to train a honey bee. Journal of Undergraduate Neuroscience Education 17(1):T1-T11]. All wells are filled at the start of the observation period, with each well containing approximately 375 µl of sucrose solution. This citation, which includes technical details, was already included in the paper. We added the volume of sucrose solution in each well to Line 209.

Line 260-263: We recorded the ambient temperature on each outdoor testing day. During the period these experiments were conducted the temperature ranged between a low of 25˚C to a high of 28.9˚ C. Although it is formally possible that temperature significantly impacted viscosity and bee physiology, we believe that the effects are likely to be negligible given that, over this temperature range, the density of water changes from 0.996 g/cm3 to 0.997 g/cm3. It is known that temperatures of body compartments in flying honey bees change with ambient temperature [Roberts SP, Harrison JF. (1999) Mechanisms of thermal stability during flight in the honeybee Apis mellifera. Journal of Experimental Biology 202:1523-33] but the changes over the temperature range we tested in are likely to be small. We have added information about the test temperature to the manuscript (Lines 283-287).

Line 341. As suggested by Reviewer 2, we have added a brief description of the Low, Medium and High labels to the figure caption so that the reader does not need to refer back to the Methods section. This information has been added to Lines 357-359.

Lines 351- 354. We concur with Reviewer 2 that “micronutrients in nectar also greatly affect subjective evaluation of nectar, even when sugar concentrations are equal.” This is an interesting topic for further study.

Line 398. In reference to the oldest honey bees tested in our study (hive-reared 35-day-old adult workers), Reviewer 2 pointed us to an interesting paper on age-related differences in conditioning of the proboscis extension response has been studied some, for example: Laloi, D., Gallois, M., Roger, B., Pham-Delègue, M.H., (2001) Changes with age in olfactory conditioning performance of cage-rearedworker honey bees (Apis mellifera). Apidologie 32: 231-242. Although the oldest honey bees tested in Laloi et al. study were 20-day-old-adult workers, we were interested to learn that the only age-based differences were resistance to extinction (not pertinent to this study) and spontaneous responses to the presentation of odorants, which were significantly higher in the youngest honey bees tested and non-existent in 20-day-olds. This speaks to our speculation that spontaneous responses to odorants may change over the course of a worker’s life, and we have included the citation in the revised manuscript (Lines 418-419). We used the ”Articles Citing this Article” function at the Apidologie.com website but did not recover any additional pertinent articles.

Line 402. Reviewer 2 notes that most studies of sucrose sensitivity use lower concentrations. We have reported our test conditions accurately and consistently in this article and look forward to expanding the range of concentrations used in future studies.

Journal policy comment on “data not shown” (line 255). These observations were anecdotal and made as we tested the implementation of procedures described in Van Nest BN, Moore D. (2018) How to train a honey bee. Journal of Undergraduate Neuroscience Education 17(1):T1-T11. Because our goal was always to test drinking duration in the field using a low but acceptable concentration of sucrose (0.3 mol L-1, 10%) that had been previously used as an intermediate value by other investigators, particularly Mujagic and Erber (2009). We have revised the description (Lines 263-265) in the methods section to remove the inappropriate reference to unanalyzed pilot data not shown as we based our choice of sucrose concentration for these studies on the published literature. In addition, as requested, we have provided captions for the supporting information files and checked the references.

We thank the Reviewers for their work. It is our hope that this manuscript is now acceptable for publication in PLoS One.

---

## [Decision Letter · Decision Letter 1]

4 Dec 2023

Impact of odorants on perception of sweetness by honey bees

PONE-D-23-24418R1

Dear Dr. Fahrbach,

We’re pleased to inform you that your manuscript has been judged scientifically suitable for publication and will be formally accepted for publication once it meets all outstanding technical requirements.

Kind regards,

Wolfgang Blenau

Academic Editor

PLOS ONE

Additional Editor Comments (optional):

Reviewers' comments:

Reviewer's Responses to Questions

**Comments to the Author**

1. If the authors have adequately addressed your comments raised in a previous round of review and you feel that this manuscript is now acceptable for publication, you may indicate that here to bypass the “Comments to the Author” section, enter your conflict of interest statement in the “Confidential to Editor” section, and submit your "Accept" recommendation.

Reviewer #1: All comments have been addressed

Reviewer #2: All comments have been addressed

2. Is the manuscript technically sound, and do the data support the conclusions?

Reviewer #1: Yes

Reviewer #2: Yes

3. Has the statistical analysis been performed appropriately and rigorously? 

Reviewer #1: Yes

Reviewer #2: Yes

4. Have the authors made all data underlying the findings in their manuscript fully available?

Reviewer #1: Yes

Reviewer #2: Yes

5. Is the manuscript presented in an intelligible fashion and written in standard English?

Reviewer #1: Yes

Reviewer #2: Yes

6. Review Comments to the Author

Reviewer #1: I am pleased with the review of the manuscript. In my opinion the authors took all the comments by the reviewers seriously, and change their manuscript appropriately.

Reviewer #2: The authors responded to all reviewers’ comments. The paper is ready for publication as is. No further comments.

7. PLOS authors have the option to publish the peer review history of their article (what does this mean?). If published, this will include your full peer review and any attached files.

Reviewer #1: No

Reviewer #2: **Yes: **Sharoni Shafir

---

## [Editor Report · Acceptance letter]

6 Dec 2023

PONE-D-23-24418R1 

Impact of odorants on perception of sweetness by honey bees 

Dear Dr. Fahrbach:

I'm pleased to inform you that your manuscript has been deemed suitable for publication in PLOS ONE. Congratulations! Your manuscript is now with our production department. 

Kind regards, 

on behalf of

Dr. Wolfgang Blenau 

Academic Editor

PLOS ONE